# CV-VAE: A Compatible Video VAE for Latent Generative Video Models

**Sijie Zhao**    **Yong Zhang** ✉    **Xiaodong Cun**    **Shaoshu Yang**    **Muyao Niu**

**Xiaoyu Li**    **Wenbo Hu**    **Ying Shan**

Tencent AI Lab

`https://github.com/AILab-CVC/CV-VAE`

## Abstract

Spatio-temporal compression of videos, utilizing networks such as Variational Autoencoders (VAE), plays a crucial role in OpenAI's SORA and numerous other video generative models. For instance, many LLM-like video models learn the distribution of discrete tokens derived from 3D VAEs within the VQVAE framework, while most diffusion-based video models capture the distribution of continuous latent extracted by 2D VAEs without quantization. The temporal compression is simply realized by uniform frame sampling which results in unsmooth motion between consecutive frames. Currently, there lacks of a commonly used continuous video (3D) VAE for latent diffusion-based video models in the research community. Moreover, since current diffusion-based approaches are often implemented using pre-trained text-to-image (T2I) models, directly training a video VAE without considering the compatibility with existing T2I models will result in a latent space gap between them, which will take huge computational resources for training to bridge the gap even with the T2I models as initialization. To address this issue, we propose a method for training a video VAE of latent video models, namely CV-VAE, whose latent space is compatible with that of a given image VAE, *e.g.,* image VAE of Stable Diffusion (SD). The compatibility is achieved by the proposed novel latent space regularization, which involves formulating a regularization loss using the image VAE. Benefiting from the latent space compatibility, video models can be trained seamlessly from pre-trained T2I or video models in a truly spatio-temporally compressed latent space, rather than simply sampling video frames at equal intervals. To improve the training efficiency, we also design a novel architecture for the video VAE. With our CV-VAE, existing video models can generate four times more frames with minimal finetuning. Extensive experiments are conducted to demonstrate the effectiveness of the proposed video VAE.

## 1 Introduction

Video generation has gained significant public attention, especially after the announcement of OpenAI SORA [2]. Current popular video models can be divided into two categories based on the modeling space, *i.e.,* pixel and latent space. Imagen Video [17], Make-a-video [29], and Show-1 [42] are representative video diffusion models that directly learn the distribution of pixels. On the other hand, Phenaki [33], MAGVIT [41], VideoCrafter [6], AnimateDiff [15], VideoPeot [20], and SORA, *etc*, are representative latent generative video models that are trained in the latent space formed using variational autoencoders (VAEs). The latter category is more prevalent due to its training efficiency.

---

✉Corresponding author

38th Conference on Neural Information Processing Systems (NeurIPS 2024).

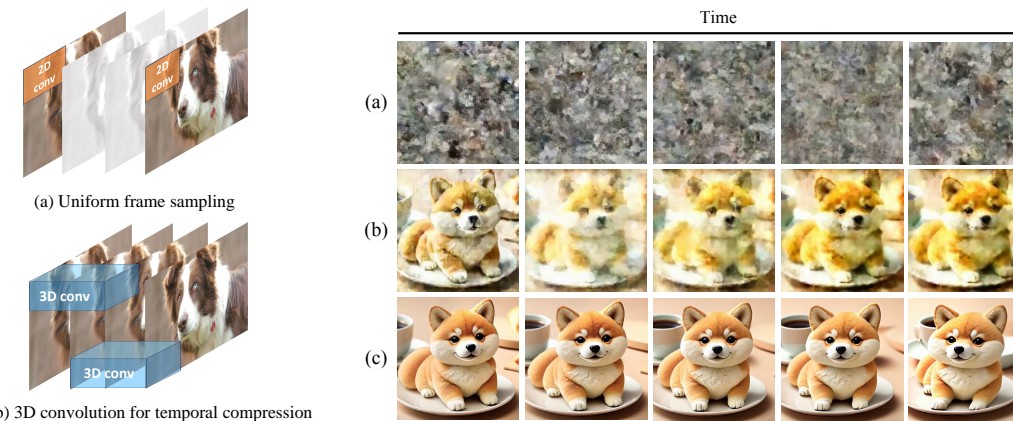

Figure 1: Temporal compression difference between an image VAE and our video one.

(a) Uniform frame sampling

(b) 3D convolution for temporal compression

Figure 2: SVD inference (a) The pretrained SVD with the independently trained Video VAE. (b) Finetuning SVD based on (a). (c) The pretrained SVD with our video VAE.

Furthermore, latent video generative models can be classified into two groups according to the type of VAE they utilize: LLM-like and diffusion-based video models. LLM-like models train a transformer on discrete tokens extracted by a 3D VAE with a quantizer within the VQ-VAE framework [32]. For example, VideoGPT [40] initially trains a 3D-VQVAE and subsequently an autoregressive transformer in the latent space. The 3D-VQVAE is inflated from the 2D-VQVAE [32] used in image generation. TATS [14] and MAGVIT [41] use 3D-VQGAN for better visual quality by employing discriminators, while Phenaki [33] utilizes a transformer-based encoder and decoder, namely CViViT.

However, recent latent diffusion-based video models typically exploit 2D VAEs, rather than 3D VAEs, to generate continuous latents to train a UNet or DiT [25]. The commonly used 2D VAE is the image VAE [28] from Stable Diffusion, as training a video model from scratch can be quite challenging. Almost all high-performing latent video models are trained with the SD image model [28] as initialization for the inflated UNet or DiT. Examples include Align-your-latent [5], VideoCrafter1 [6], AnimateDiff [15], SVD [4], Modelscope [34], LaVie [35], MagicVideo [44], Latte [24], *etc*. Temporal compression is simply achieved by uniform frame sampling while ignoring the motion information between frames (see Fig. 1). Consequently, the trained video models may not fully understand smooth motion, even when FPS is set as a condition. When projecting a sampled latent sequence to a video using the decoder of the 2D VAE, the generated video exhibits a low FPS and lacks visual smoothness.

Currently, the research community lacks a commonly used 3D video VAE for generating continuous latent variables with spatio-temporal compression for latent video models. Training a high-quality video VAE without considering the compatibility with existing pretrained image and video models might not be too difficult. However, even though the trained video VAE exhibits low reconstruction errors, a gap exists between its learned latent space and the one used by pretrained models, such as the video VAE of Open-Sora-Plan [1]. This means that bridging the gap requires significant computational resources and extensive training time, even when using pre-trained models as initialization. One example is shown in Fig. 2. When training a video VAE independently without considering compatibility, the sampled latent of SVD [4] cannot be projected into the pixel space correctly due to the latent space gap, as shown in Fig. 2(a). After finetuning the SVD model in the new latent space on 16 A100 for 58K iterations, the quality of the generated video is still poor (see Fig. 2(b)). In contrast, our video VAE achieves promising results in the pretrained SVD even without finetuning the UNet as shown in Fig. 2(c).

In this work, we propose a novel method to train a video VAE to extract continuous latents for generative video models, which is compatible with existing pretrained image and video models, *e.g.* Stable Diffusion [28] and SVD [4]. We also inflate the SD image VAE to form a video VAE by adding 3D convolutions to both encoder and decoder of the 2D VAE, which allows us to train video models efficiently with the pretrained models as initialization in a truly spatio-temporally compressed

latent space, instead of uniform frame sampling for temporal compression (see Fig. 1). Consequently, the generated videos will be smoother and have a higher FPS than those produced using a 2D VAE.

To ensure latent space compatibility between 2D and 3D VAEs, we propose a latent space regularization to avoid distribution shifts. We examine the effectiveness of using either the encoder or decoder of the 2D VAE to form constraints and explore four types of mapping functions to design regularization. Moreover, to improve video VAE efficiency, we investigate its architecture and partially integrate 3D convolutions instead of exploiting 3D convolution in all blocks. The proposed video VAE can be used not only for training new video models with pretrained ones as initialization but also as a frame interpolator for existing video models with slight finetuning.

Our main contributions are summarized as follows: (1) We propose a video VAE that provides a truly spatio-temporally compressed continuous space for training latent generative video models, which is compatible with existing image and video models, greatly reducing the expense of training or finetuning video models. (2) We propose a latent space regularization to avoid distribution shifts and design an efficient architecture for the video VAE. (3) Extensive experiments are conducted to demonstrate the effectiveness of the proposed video VAE.

## 2   Related Work

**Variational Autoencoder.**   Variational Autoencoders (VAEs), introduced by [19], have been widely used in two-stage generative models. The first stage involves compressing the pixels into a lower-dimensional latent representation, followed by a second stage that generates pixels from this latent space. VAEs can be divided into two groups according to the token, *i.e.,* discrete and continuous latent. The difference between the two types of VAEs is the quantization. Continuous VAEs have no quantization, while discrete VAEs learn a codebook for quantization and use it to convert the continuous latent features to discrete indices, called VQVAE [32]. When training discrete VAEs, some methods exploit a discriminator to improve the visual image quality, called VQGAN [12].

In video generation, 2D VAEs are typically inflated into 3D ones by injecting 3D Conv or temporal attention. 3D Convs are for CNN-based VAEs, *e.g.,* 3D-VQVAE [40], 3D-VAQGAN [14, 41]. Attentions are for transformer-based VAEs, *e.g.,* CViViT [33]. Although there are several discrete 3D VAEs for video generation, there are no commonly used continuous 3D VAEs.

**Video Generative models.**   Video generation has achieved remarkable progress in recent years. The announcement of Imagen Video [17] and Make-A-Video [29] made researchers see the hope of purely AI-generated videos. Then, the launch of OpenAI SORA [2] brought the enthusiasm of researchers in academia and industry to a climax. Many video generation models [17, 29, 42] directly learn the distribution of pixels while some others [5, 6, 44, 34, 35, 24, 41, 14, 33, 40, 4, 15] learn the distribution of tokens in a latent space. The tokens are always extracted by a variational autoencoder [11]. Latent video generation models can be categorized into two groups according to whether the token is discrete or continuous. TATS [14], MAGVIT [41], VideoGPT [40], and Phenaki [33] are representative models trained with discrete tokens extracted by a 3D VAE within the VQVAE framework [32]. A codebook is learned jointly with the VAE for quantization. SVD [4], AnimateDiff [15], VideoCrafter [6], *etc.*, are video models trained with continuous latent extracted by a 2D VAE without quantization, rather than a 3D VAE. SD image VAE is the commonly used 2D VAE. One reason is that video models are difficult to train from scratch and they are always initialized with the weights of a pretrained T2I model such as Stable Diffusion UNet [28]. Hence, the corresponding image VAE is used to extract latents from a video. Since the image VAE can only perform spatial compression, the temporal compression is realized by uniform frame sampling. This strategy ignores the motion between key frames.

There lacks a video VAE that is compatible with the pretrained T2I or video models. Though it is not difficult to train a video VAE (3D VAE) independently with high reconstruction accuracy; it will result in a latent space gap between the learned video VAE and existing pre-trained image and video models that are always used as initialization. The Open-Sora-Plan project [1] offers a video VAE; however, it is not compatible with existing image or video models. Large computational resources and a long training time are required to bridge the gap. In this work, we propose a latent space regularization method to train a video VAE whose latent space is compatible with pretrained models.

## 3 Method

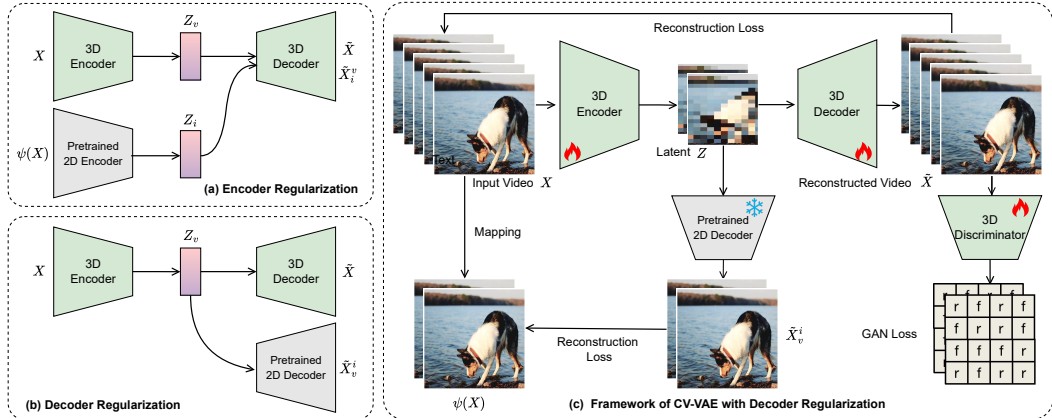

Figure 3: (a-b): Two different regularization methods; (c) The framework of CV-VAE with the regularization of the pretrained 2D decoder.

We propose a latent space regularization method for training a video VAE that is compatible with pre-trained image and video models. We examine multiple strategies for implementing the regularization, focusing on either the encoder or the decoder of the image VAE. Additionally, we explore four types of mapping functions to develop the regularization loss. To enhance the efficiency of the video VAE, we introduce an architecture that employs different inflation strategies in distinct blocks, instead of incorporating 3D convolutions in all blocks.

### 3.1 Latent Space Regularization

We inflate a 2D VAE into a 3D VAE, initializing it with the 2D VAE's weights. The 3D VAE is designed to be capable of encoding both image and video (see details in Sec.3.2). The key of building a compatible video VAE is the latent space alignment between the video VAE and the image VAE.

**Notations.** Let $x \in \mathbb{R}^{H \times W \times 3}$ denote an image in RGB space and $X \in \mathbb{R}^{(T+1) \times H \times W \times 3}$ denote a video with $T + 1$ frames. When $T = 0$, $X$ degrades into an image and the video VAE will process it with temporal padding. $z \in \mathbb{R}^{h \times w \times c}$ denotes the latent tokens extracted by either an image VAE or a video VAE. $Z \in \mathbb{R}^{(t+1) \times h \times w \times c}$ is latent tokens extracted by the video VAE. $\rho_s = H/h = W/w$ and $\rho_t = T/t$ are the spatial and temporal compression rates. Let $\mathcal{E}_i$ and $\mathcal{D}_i$ denote the encoder and decoder of the image VAE, respectively. While $\mathcal{E}_v$ and $\mathcal{D}_v$ are for the video VAE. Then, we have $z = \mathcal{E}_i(x)$, $Z = \mathcal{E}_v(X)$, and $z = \mathcal{E}_v(x)$. $\tilde{x} = \mathcal{D}_i(z) = \mathcal{D}_i(\mathcal{E}_i(x))$, $\tilde{X} = \mathcal{D}_v(Z) = \mathcal{D}_v(\mathcal{E}_v(X))$, and $\tilde{x} = \mathcal{D}_v(z) = \mathcal{D}_v(\mathcal{E}_v(x))$ are the reconstructed image and video from the latent tokens.

**Regularization.** We assume the latent of the image VAE follow a distribution, *i.e,* $z \sim p^i(z)$. The joint distribution of $t + 1$ independent frames is $p^i(Z) = \prod_k^{t+1} p^i(z_k)$. The latent distribution of the video VAE can be denoted as $Z \sim p^v(Z)$. To achieve the alignment between the latent spaces of the image and video VAEs, we have to build mappings between $p^i(Z)$ and $p^v(Z)$. Since both distributions have no analytic formulation, distance metric for measuring differences between distributions is not applicable.

Here, we build the cooperation between the image VAE and the video one to construct reconstruction loss for space alignment. When exploiting the encoder of the image VAE for alignment, the latent extracted from the image encoder should be corrected decoded by the decoder of the video VAE, *i.e.*, $\tilde{X}_i^v = \mathcal{D}_v(\mathcal{E}_i(\psi(X)))$. The illustration is shown in Fig. 3(a). For a given input video $X \in \mathbb{R}^{(T+1) \times H \times W \times 3}$, we use a mapping function $\psi$ to sample $\psi(X) \in \mathbb{R}^{(T/\rho_t+1) \times H \times W \times 3}$. Thus the reconstructed video $\tilde{X}_i^v$ is the same as the shape of $X$. Then, the reconstruction loss of using the image encoder can be defined as

$$L_{\text{reg}}^{\text{en}} = ||X - \tilde{X}_i^v||^2. \tag{1}$$

When exploiting the decoder of the image VAE, the latent extracted by the video encoder can be decoded by the decoder of the image VAE, *i.e.*, $\tilde{X}_v^i = \mathcal{D}_i(\mathcal{E}_v(X))$. The illustration is shown in Fig. 3(b). For a given input video $X \in \mathbb{R}^{(T+1)\times H\times W\times 3}$, the reconstructed video is $\tilde{X}_i^v \in \mathbb{R}^{(T/\rho_t+1)\times H\times W\times 3}$. Then, the reconstruction loss of using the image decoder can be defined as

$$L_{\text{reg}}^{\text{dec}} = ||\psi(X) - \tilde{X}_v^i||^2. \tag{2}$$

**Mapping Functions.**    To bridge the dimension gap between $\tilde{X}_v^i$ or $\tilde{X}_i^v$ and $X$, we investigate four types of mapping functions $\psi$ as follows. **1) First frame.** We compare only the first frame of the input video and the reconstructed one. The regularization loss degenerates to measure the difference between the input and reconstruction of the image. **2) Slice.** $\psi$ samples one frame every $\rho_t$ frames to form a shorter video. It starts from the second frame and the first one is reserved. **3) Average.** $\psi$ computes the average of every $\rho_t$ frames, starting from the second frame. **4) Random.** $\psi$ randomly samples one frame from every $\rho_t$ frames, starting from the second frame.

**Training Objective.**    Following the training of the 2D VAE in LDM [28], our basic objective is a combination of a reconstruction loss [43], an adversarial loss [12], and a KL regularization [19], *i.e.*,

$$L_{\text{AE}} = \min_{\mathcal{E}_v, \mathcal{D}_v} \max_{D_v} \; L_{\text{rec}}(X, \mathcal{D}_v(\mathcal{E}_v(X)) - L_{\text{adv}}(\mathcal{D}_v(\mathcal{E}_v(X))) + \log D_v(X) + L_{\text{KL}}(X; \mathcal{E}_v, \mathcal{D}_v),$$

where the first term is the reconstruction loss, the second and third are the adversarial loss, and the last is the KL regularization. $D_v$ is the discriminator that differentiates original videos from reconstructed ones. It is inflated from the image discriminator in LDM by injecting 3D convolutions. Then, for latent space alignment, our full training objective is:

$$L_{\text{AE}}^{\text{align}} = L_{\text{AE}} + \lambda_1 L_{\text{reg}}^{\text{dec}} + \lambda_2 L_{\text{reg}}^{\text{en}}, \tag{3}$$

where $\lambda_1$ and $\lambda_2$ are trade-off parameters. We explore different settings of $\lambda_1$ and $\lambda_2$ and find that using the decoder only achieves the best performance. The framework of CV-VAE is shown in Fig. 3(c) and evaluations between different regularization methods can be found in Tab. 6.

## 3.2    Architecture Design of Video VAE

We design the architecture of the video VAE according to the image VAE in LDM [28]. The detailed architecture is presented in the Appendix A.1. We explain the key modifications as follows.

**Model Inflation.**    Considering the latent space compatibility and the convergence speed of the video VAE, we make full use of the pretrained weights of the image VAE for initialization, instead of training from scratch. We inflate the image VAE into the video VAE by replacing 2D convolutions with 3D convolutions. 3D convolutions are used to model the temporal dynamics among frames. To initialize the 3D convolutions, we copy the weights of the 2D Conv kernel to the corresponding positions in the 3D Conv kernel and set the remaining parameters to zero. We set the stride to achieve temporal downsampling and increase the number of 3D kernels by a factor of $s$ to achieve $s\times$ temporal upsampling. To enable the video VAE to handle both image and video, given $T+1$ frames as input, we use reflection padding in the temporal dimension for the first frame. By initializing the video VAE using the above operations, we can reconstruct images without training, significantly accelerating the training convergence speed on video datasets.

**Efficient 3D Architecture.**    Expanding 2D Convs to 3D Convs (e.g., $k\times k \to k\times k\times k$) results in $k\times$ parameters and computational complexity. To improve the computational efficiency of the model, we adopt a 2D+3D network structure. Specifically, we retain half of the convolutions in the ResBlock as 2D Convs and set the other half as 3D Convs. We find that, compared to setting all Convs to 3D, the number of parameters and the computational complexity are reduced by roughly 30%, while the reconstruction performance remains nearly the same. See Sec. 4.2 for experimental comparisons.

**Temporal Tiling for Arbitrary Video Length**    Existing image VAEs employ spatial tiling on large spatial resolution images to achieve memory-friendly processing, which cannot handle long videos. As a result, we introduce temporal tiling processing. During encoding, the video $X$ is divided into $[X_1, X_2, ...X_n]$, where $X_i \in \mathbb{R}^{(1+f\cdot\rho_t)\times H\times W\times 3}$ and $f$ is a parameter controlling the size of each

| Method | Params | FCR | Comp. | COCO-Val | | | Webvid-Val | | |
|--------|--------|-----|-------|----------|---|---|------------|---|---|
| | | | | PNSR($\uparrow$) | SSIM($\uparrow$) | LPIPS($\downarrow$) | PNSR($\uparrow$) | SSIM($\uparrow$) | LPIPS($\downarrow$) |
| VAE-SD2.1 [28] | 34M + 49M | 1x | - | 26.6 | 0.773 | **0.127** | **28.9** | 0.810 | 0.145 |
| VQGAN [12] | 26M + 38M | 1x | × | 22.7 | 0.678 | 0.186 | 24.6 | 0.718 | 0.179 |
| TATS [14] | 7M + 16M | 4x | × | 23.4 | 0.741 | 0.287 | 24.1 | 0.729 | 0.310 |
| VAE-OSP [1] | 94M +135M | 4x | × | 27.0 | 0.791 | 0.142 | 26.7 | 0.781 | 0.166 |
| Ours(2D+3D) | 68M + 114M | 4x | ✓ | 27.6 | **0.805** | 0.136 | 28.5 | 0.817 | **0.143** |
| Ours(3D) | 100M + 156M | 4x | ✓ | **27.7** | **0.805** | 0.135 | 28.6 | **0.819** | 0.145 |

Table 1: Quantitative evaluation on image and video reconstruction. FCR represents the frame compression rate, and Comp. indicates compatibility with existing generative models.

block. $X_i$ and $X_{i+1}$ have a one-frame overlap in the temporal dimension. After encoding each $X_i$ to obtain $Z_i$, we discard the first frame of $Z_i$ when $i \neq 0$ and concatenate all $Z_i$ in the temporal dimension to obtain $Z$. The decoding process is handled similarly to the encoding process. By combining our method with 2D tiling, we can encode videos with arbitrary resolution and length.

## 4 Experiments

### 4.1 Experimental Setups

**Datasets and Metrics.** We evaluate our CV-VAE on the COCO2017 [21] validation dataset and the Webvid [3] validation dataset which includes 1024 videos. Both images and videos are resized and cropped to a resolution of $256 \times 256$. Each video is sampled with 33 frames and a frame stride of 3. We evaluate the reconstruction performance of CV-VAE on images and videos using metrics such as PSNR, SSIM [36], and LPIPS scores [43]. We employ 3D tiled processing to encode and decode videos with arbitrary resolution and length within a limited memory footprint. During inference, we allow a single video block size of $17 \times 576 \times 576$. We evaluate the video generation quality of our model using 2048 randomly sampled videos from UCF101 [30] and MSR-VTT [39]. Videos are resized and cropped to a resolution of $576 \times 1024$ to fit the SVD [4]. We use Frechet Video Distance (FVD) [31], Kernel Video Distance (KVD) [31], and Perceptual Input Conformity (PIC) [38] metrics to evaluate video generation quality. For evaluating image generation quality, we use 2048 samples from the COCO2017 validation dataset and employ FID [16], CLIP score [26], and PIC score metrics.

**Training Details.** We train our CV-VAE model using image datasets including LAION-COCO [9] and Unsplash [23], as well as the video dataset Webvid-10M [3]. For image datasets, we employ two resolutions, *i.e.*, $256 \times 256$ and $512 \times 512$. In the case of video datasets, we use two settings of frames and resolutions: $9 \times 256 \times 256$ and $17 \times 192 \times 192$. The batch sizes for these four settings are 8, 2, 1, and 1, with sampling ratios of 40%, 10%, 25%, and 25%, respectively. We employed the AdamW optimizer [22] with a learning rate of 1e-4 and cosine learning rate decay. To avoid numerical overflow, we trained CV-VAE using float32 precision, and the training was carried out on 16 A100 GPUs for 200K steps. To fine-tune the SVD on CV-VAE, we utilize in-house data with a frame rate and resolution of $97 \times 576 \times 1024$. We employ deepspeed stage 2 [27], gradient checkpointing [8] techniques, and train with bfloat16 precision. We used a constant learning rate of 1e-5 with the AdamW [22] optimizer, and only optimized the last layer of U-Net. The training was carried out on 16 A100 GPUs for 5K steps.

### 4.2 Image and Video Reconstruction

We evaluated the reconstruction quality of various VAE models on image and video test sets. The comparison group includes: (1) VAE-SD2.1 [28] which is widely used in the community for image and video generation models. (2) VQGAN [12] which encoding pixels into discrete latents. We use the f8-8192 version for comparision. (3) TATS [14]: a 3D VQGAN designed for video generation. (4) VAE-OSP [1]: a 3D VAE from Open-Sora-Plan which is initialized from VAE-SD2.1 and trained with video data. (5) Our CV-VAE (2D+3D): retains half of the 2D convolutions to reduce computational overhead. (6) Our CV-VAE (3D): utilizes only 3D convolutions.

As illustrated in Tab. 1, we present the parameter count (Params), Frame Compression Ratio (FCR), and compatibility with existing diffusion models (Comp.) for various VAE models. Thanks to the latent constraint, our model is compatible with current diffusion models, compresses videos by $4\times$ in the temporal dimension, and achieves top-tier image and video reconstruction quality. This enables the generation of longer videos under roughly the same computational resources. Reconstruction

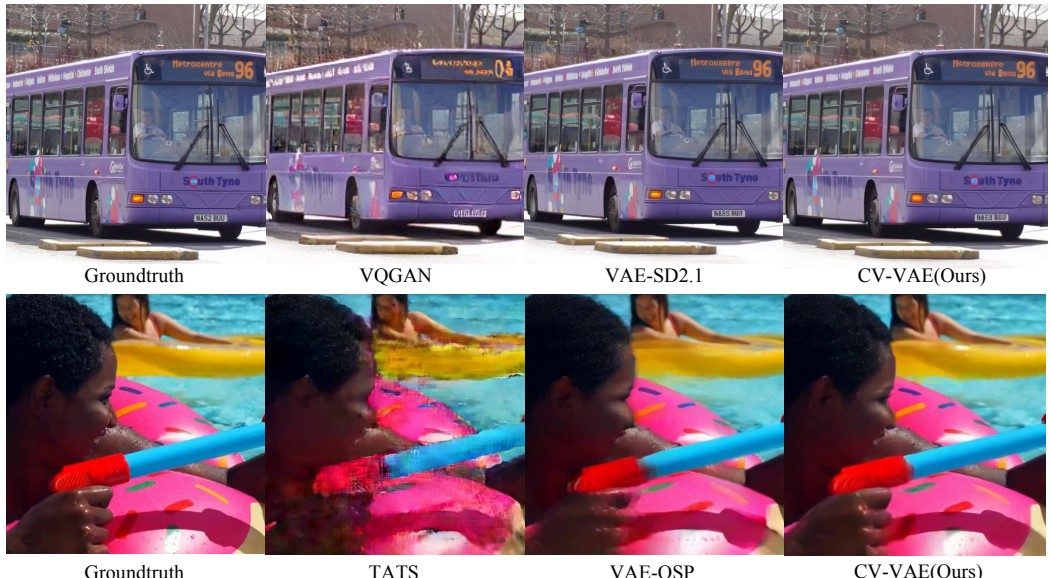

Figure 4: Qualitative comparison of image and video reconstruction. Top: Reconstruction with different Image VAE models (*i.e.*, VQGAN [12] and VAE-SD2.1 [28] ) on images; Bottom: Reconstruction with different Video VAE models (*i.e.*, TATS [14] and VAE-OSP [1]) on video frames.

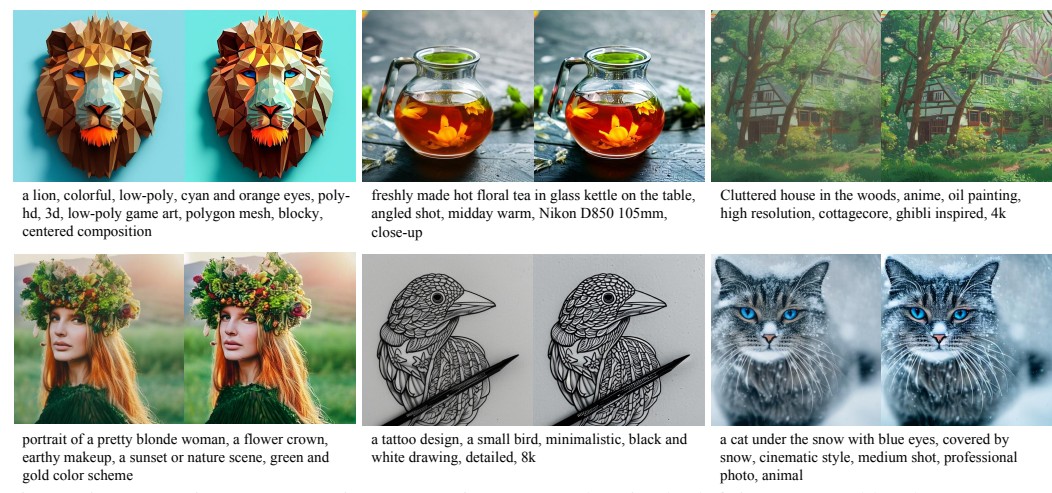

a lion, colorful, low-poly, cyan and orange eyes, poly-hd, 3d, low-poly game art, polygon mesh, blocky, centered composition

freshly made hot floral tea in glass kettle on the table, angled shot, midday warm, Nikon D850 105mm, close-up

Cluttered house in the woods, anime, oil painting, high resolution, cottagecore, ghibli inspired, 4k

portrait of a pretty blonde woman, a flower crown, earthy makeup, a sunset or nature scene, green and gold color scheme

a tattoo design, a small bird, minimalistic, black and white drawing, detailed, 8k

a cat under the snow with blue eyes, covered by snow, cinematic style, medium shot, professional photo, animal

Figure 5: Text-to-image generation comparison. In each pair, the left is generated by the SD2.1 [28] with the image VAE while the right is generated by the SD2.1 with our video VAE.

quality improves as the number of latent channels increases. For comparison results with 16 latent channels, please refer to Appendx A.2.

We also conducted a qualitative comparison of the reconstruction results for different VAE models, as shown in Fig. 4. In the top row, we reconstructed images with a resolution of $512 \times 512$ and compared them with Image VAE models. All three models compressed the images to a latent size of $64 \times 64$. Our results were close to those of VAE-SD2.1, while VQGAN had the worst performance. In the bottom row, we reconstructed videos with a resolution of $33 \times 512 \times 512$ and compared them with Video VAE models. All three models compressed the videos to a latent size of $9 \times 64 \times 64$. Comparing the decoded videos at the same frames, our model achieved the best results. Check Appendx A.3 and A.4 for more reconstruction results.

### 4.3 Compatibility with Existing Models

**Text-to-Image Models**     We tested the compatibility of our CV-VAE by integrating it into the pretrained SD2.1 [28], replacing the original 2D VAE without any finetuning. We evaluated it on

|  | Trainable | COCO2017-Val | | |
|---|---|---|---|---|
|  |  | FID($\downarrow$) | CLIP($\downarrow$) | PIC($\uparrow$) |
| SD2.1 [28] | $\times$ | 57.3 | 0.312 | 0.354 |
| SD2.1+CV-VAE | $\times$ | 57.6 | 0.311 | 0.360 |

Table 2: Quantitative results of text-to-image generation.

| Method | Trainable | FCR | Frames | UCF-101 | | | MSR-VTT | | |
|---|---|---|---|---|---|---|---|---|---|
|  |  |  |  | FVD($\downarrow$) | KVD($\downarrow$) | PIC($\uparrow$) | FVD($\downarrow$) | KVD($\downarrow$) | PIC($\uparrow$) |
| SVD [4] | $\times$ | 1$\times$ | 25 | **402** | 8.20 | 0.791 | 310 | **1.30** | 0.588 |
| SVD+CV-VAE-I | $\times$ | 1$\times$ | 25 | 419 | **6.73** | 0.763 | **262** | 1.67 | 0.609 |
| SVD+CV-VAE-V | $\times$ | 4$\times$ | 97 | 762 | 15.7 | 0.791 | 319 | 3.31 | 0.696 |
| SVD+CV-VAE-V | Output layer | 4$\times$ | 97 | **681** | **13.1** | **0.858** | **295** | **2.26** | **0.734** |

Table 3: Evaluation results of image-to-video generation. FCR denotes the frame compression rate.

the COCO-Val [21] dataset and compared the results with the SD2.1 model using PID, CLIP score, and PIC metrics. The data (see Tab. 2) suggest that both models perform similarly in text-to-image generation.

We also visualized the text-to-image generation results of both models in Fig. 5. In each pair, the left side depicts the results of SD2.1, while the right side shows the results generated by our CV-VAE, which replaced the original VAE, using the same random seed and prompt. The results show that both models generate images with almost identical content and texture, with only slight differences in color. This further validates the feasibility of building a compatible VAE via latent constraint.

**Image-to-Video Models**   The primary objective of CV-VAE is to train a model that can compress both time and space, while also being compatible with the existing 2D VAE. In this section, we validate the compatibility of CV-VAE with existing video generation models. We integrate CV-VAE into SVD [4], replacing the original VAE, and decoded the generated video latents. CV-VAE offers the flexibility to decode either in image mode (CV-VAE-I) or video mode (CV-VAE-V); the former decodes n frames of latent into n frames of video, while the latter decodes n frames of latent into $1 + (n - 1) \times 4$ frames of video. We tested the video generation quality of both models. Furthermore, we fine-tuning the SVD for better alignment.

As shown in Tab. 3, incorporating 'CV-VAE-I' into a frozen SVD immediately yields video generation quality comparable to the original VAE. Using CV-VAE in video mode can also decode videos generated by SVD, and further improvements in video decoding quality can be achieved by fine-tuning only the output layer (approximately 12k parameters). One of the reasons for the noticeable gap in test metrics between 'SVD+CV-VAE-I' and 'SVD+CV-VAE-V' is that they use different numbers of frames, making a direct comparison challenging.

In Fig. 6, we also display the comparison results with SVD [4]. The top row shows the generated results by SVD, and the bottom row shows the generated results after inserting CV-VAE into SVD and fine-tuning the output layer. We use the first frame as a condition and generate with the same random seed. The U-Net generates 25 frames of latent, which are decoded by CV-VAE into a 97-frame video. As can be seen, compared to the original SVD, our results exhibit smoother motion. It is worth noting that both models have the same computational complexity during the diffusion process, which means that our model is more scalable.

By fine-tuning a small number of parameters, the image-to-video model can generate smoother and longer videos through CV-VAE, effectively serving as a frame interpolation method. Therefore, we compared CV-VAE with existing video interpolation models [18], conducting experiments on MSR-VTT [39]. We first used SVD to generate a video of size $25 \times 576 \times 1024$, and then applied the interpolation model to expand the video from 25 frames to 97 frames. For our results, we directly generated a video of size $97 \times 576 \times 1024$ using CV-VAE and the fine-tuned SVD. The comparison results are shown in Tab. 4.3, where CV-VAE outperformed the FIRE [18] in two out of three metric, validating the potential of CV-VAE as an interpolation model.

**Text-to-Video Models**   In this section, we integrate CV-VAE into the existing text-to-video model to futher validate the effectiveness. 'VC2' refers to decoding the results generated by VideoCrafter2 [7] using the original 2D VAE, 'VC2+CV-VAE-I' indicates decoding the results using CV-VAE in image mode, and 'VC2+CV-VAE-V' denotes decoding the results using CV-VAE in video mode, which generates videos of 4$\times$ frames. We only fine-tuned a small number of parameters in U-Net, including

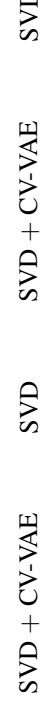

SVD

SVD + CV-VAE

SVD

SVD + CV-VAE

Figure 6: Comparison between the image VAE and our video VAE on image-to-video generation of SVD [4]. 'SVD' means using the image VAE. 'SVD + CV-VAE' means using our video VAE and tuning the output layer of SVD. *Click to play the video clips with Adobe or Foxit PDF Reader.*

| Method | FVD($\downarrow$) | KVD($\downarrow$) | PIC($\uparrow$) |
|---|---|---|---|
| SVD+RIFE [18] | **253** | 3.12 | 0.721 |
| SVD+CV-VAE | 295 | **2.26** | **0.734** |

Table 4: Comparison between CV-VAE and frame interpolation model.

the first and last layers. We use captions from the validation set of MSR-VTT [39] for evaluation, with the resolution of 320×512. Following the approach taken by previous studies [37], we used the CLIP [26] metric to evaluate the generation quality of text-to-video models, including Frame Consistency (F.C.) and Textual Alignment (T.A.). The experimental results are shown in Tab. 4.3, where the 'VC2+CV-VAE-V' setting achieved the best generation performance through fine-tuning VideoCrafter2. Check Appendx A.5 for quantitative comparison.

## 4.4 Ablation Study

**Influence of Regularization Type** We evaluated the impact of three types of latent regularization, which are: (1) **2D Enc.** , *i.e.*, $\lambda_1 = 0$ and $\lambda_2 = 1$ in Eq. 3; (2) **2D Dec.** , *i.e.*, $\lambda_1 = 1$ and $\lambda_2 = 0$ in Eq. 3; (3) **2D Enc. + Dec.** , *i.e.*, $\lambda_1 = 1$ and $\lambda_2 = 1$ in Eq. 3.

| Methods | FCR | Trainable | CLIP Metric | |
|---|---|---|---|---|
| | | | F.C. | T.A. |
| VC2 [7] | 1× | × | 0.292 | 0.960 |
| VC2+CV-VAE-I | 1× | × | 0.290 | 0.964 |
| VC2+CV-VAE-V | 4× | × | 0.285 | 0.953 |
| VC2+CV-VAE-V | 4× | ✓ | **0.301** | **0.987** |

Table 5: Evaluation results of text-to-video generation.

| Constraint | COCO-Val | | | Webvid-Val | | |
|---|---|---|---|---|---|---|
| | PNSR(↑) | SSIM(↑) | LPIPS(↓) | PNSR(↑) | SSIM(↑) | LPIPS(↓) |
| 2D Enc. | 26.0 | 0.759 | 0.205 | 26.0 | 0.748 | 0.222 |
| 2D Dec. | 27.5 | 0.801 | 0.151 | **28.0** | **0.803** | **0.158** |
| 2D Dnc. + Dec. | **27.9** | **0.808** | **0.150** | 27.6 | 0.795 | 0.176 |

Table 6: Comparison of different regularization types.

| Mapping Function | COCO-Val | | | Webvid-Val | | |
|---|---|---|---|---|---|---|
| | PNSR(↑) | SSIM(↑) | LPIPS(↓) | PNSR(↑) | SSIM(↑) | LPIPS(↓) |
| 1st Frame | 27.3 | 0.797 | 0.156 | 26.6 | 0.771 | 0.191 |
| Average | 27.5 | 0.801 | 0.151 | 27.9 | 0.801 | 0.172 |
| Slice | 27.5 | 0.802 | 0.152 | 27.7 | 0.799 | 0.168 |
| Random | **27.6** | **0.803** | **0.138** | **28.4** | **0.811** | **0.153** |

Table 7: Comparison of different mapping functions.

Tab. 6 shows the impact of various latent regularization methods. Using the 2D decoder for latent regularization results in better reconstruction for both image and video test sets compared to the 2D encoder. This is likely because the gradient backpropagation through the 2D decoder provides better guidance for the 3D VAE's learning, while the frozen 2D encoder doesn't propagate gradients. The '2D Enc. + Dec.' method performs slightly better on image test sets but worse on video datasets compared to '2D Enc.' Since our main goal is video reconstruction and for simplicity, we use the 2D decoder for regularization.

**Influence of Mapping Functions**    The 2D decoder decodes $n$ frames of latents into $n$ frames of video, while the 3D decoder decodes the same $n$ frames of latents into $1 + (n - 1) \times 4$ frames of video. Therefore, we need to mapping the input video to $n$ frames to calculate the regularization loss in Eq. 2. We evaluated four mapping functions mentioned in Sec. 3.1.

As shown in Tab. 7, the four methods have similar effects on image reconstruction, with the main differences being in video reconstruction. The '1st Frame' approach yields the worst video reconstruction results due to the lack of regularization and guidance for subsequent frames. The 'Slice' method results in poor reconstruction quality for the three unsampled middle frames. The 'Average' method is inferior to 'Random' in video reconstruction, primarily because calculating the mean for multiple consecutive frames leads to motion blur in the target.

## 5   Conclusion and Limitations

We propose a novel method to train a video VAE that is compatible with existing image and video models trained with SD image VAE. The video VAE provides a truly spatio-temporally compressed latent space for latent generative video models, as opposed to uniform frame sampling. Due to the latent space compatibility, a new video model can be trained efficiently with the pretrained image or video models as initialization. Besides, existing video models such as SVD can generate smoother videos with four times more frame using our video VAE by slightly fine-tuning a few parameters. Extensive experiments are performed to demonstrate the effectiveness of the proposed VAE.

**Limitations.**    The performance of the proposed video VAE relies on the channel dimension of the latent space. A higher dimension may yield better reconstruction accuracy. Since we pursue the latent space compatibility with existing image and video models trained with SD image VAE, the channel dimension of our video VAE is limited to be the same as the image VAE. This can be improved if an image VAE with a higher channel dimension becomes available, *e.g.,* the VAE of SD3 [13].

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

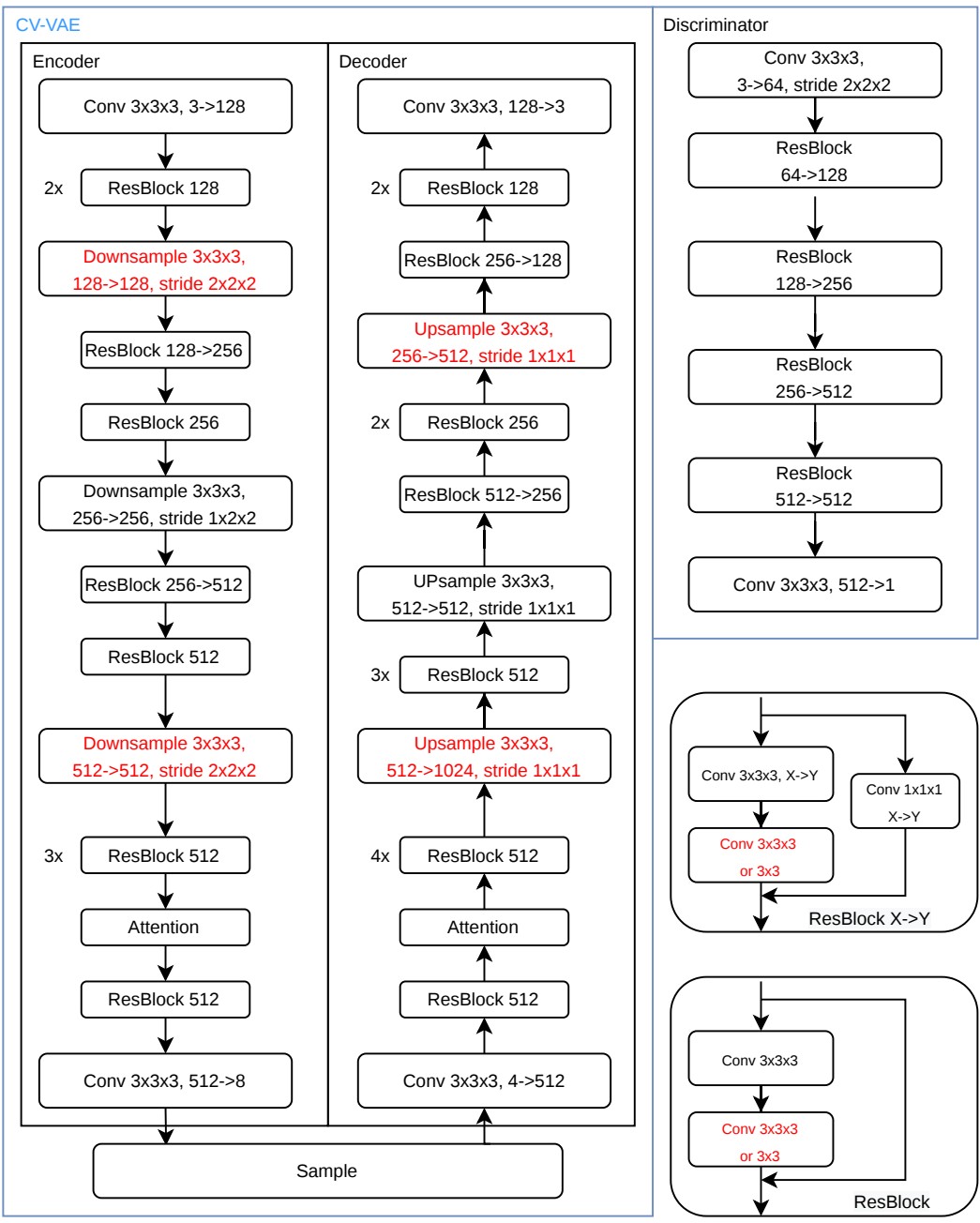

Figure 7: Architecture of CV-VAE.

# A  Appendix

## A.1  CV-VAE Model Architecture

As illustrated in Fig. 7, we introduce the structure of the CV-VAE. The architecture of CV-VAE is primarily derived from the VAE in Stable Diffusion [28], with several notable differences: (1) Some or all 2D convolutions within the network are transformed into 3D convolutions, while retaining their weights. (2) Temporal downsampling is executed in the encoder through the use of strides. (3) Temporal upsampling is accomplished by increasing the output channel number of 3D convolutions by a specific factor. (4) A discriminator, comprising 3D convolutions, is utilized. The main differences are marked in red text in Fig. 7.

| Method | Params | FCR | Comp. | COCO-Val | | | Webvid-Val | | |
|--------|--------|-----|-------|----------|--|--|------------|--|--|
| | | | | PNSR(↑) | SSIM(↑) | LPIPS(↓) | PNSR(↑) | SSIM(↑) | LPIPS(↓) |
| VAE-SD3 [13] | 34M + 49M | 1x | - | 32.6 | 0.899 | 0.064 | **33.4** | **0.910** | **0.064** |
| CVVAE-SD3 | 68M + 114M | 4x | ✓ | **33.6** | **0.916** | 0.072 | 32.0 | 0.899 | 0.087 |

Table 8: Quantitative evaluation on image and video reconstruction between. FCR represents the frame compression rate, and Comp. indicates compatibility with existing generative models.

## A.2 CV-VAE with more latent channels

More latent channels generally lead to better reconstruction performance in VAEs [13, 10], which is crucial for image and video editing tasks. In this subsection, I conducted experiments using a VAE with more latent channels (z=16). We used the 2D VAE from SD3 [13] as the baseline and trained the CVVAE-SD3 based on latent regularization. Testing was conducted under the same settings as in Tab. 1. The comparison results are shown in Tab. 8. CVVAE-SD3 outperforms VAE-SD3 in image reconstruction but is inferior to VAE-SD3 in video reconstruction. The main reason is that CVVAE-SD3 has a higher compression ratio in video compression, resulting in the loss of more information, though its reconstruction quality is significantly higher than that of VAE with z=4.

## A.3 Qualitative Examples of Image Reconstruction

In Fig. 8, we showcase additional image reconstruction results using CV-VAE. we use the version of '2D + 3D'. These images are sourced from the COCO2017 [21] dataset with a resolution of $512 \times 512$. The reconstructed image precisely shares the same colors and textures as the original, demonstrating the high fidelity of our CV-VAE in encoding and reconstructing images. Interestingly, in Fig. 5, slight color differences can be observed between the images decoded by the Image VAE and CV-VAE, given the same latent generated by the Image Diffusion Model. This suggests that there is still a minor discrepancy between the latent spaces of the video VAE trained with latent regularization and the Diffusion Model. This gap can be bridged with minimal additional training.

## A.4 Qualitative Examples of Video Reconstruction

As shown in Fig. 9, we present the reconstruction results of 4 consecutive frames from a video clip ($33 \times 576 \times 1024$) using CA-VAE. The reconstructed video frames maintain consistency in color, structure, and motion with the ground truth. According to CA-VAE, these continuous frames are condensed into a single latent frame, signifying that even a single latent frame encapsulates motion information.

## A.5 Compatibility with Existing Text-to-video Model

We tested the compatibility of CV-VAE with existing text-to-video diffusion models, such as Videocrafter2 [7], which also employs a 2D VAE from the SD as its first-stage model. We adopted a strategy similar to the training of 'SVD + CV-VAE' in Sec. 4.3, by fine-tuning the last layer of the U-Net in VC2 to adapt it to CV-VAE. We finetuned the model using in-house data at a resolution of $61 \times 320 \times 512$, which is equivalent to a latent size of $16 \times 40 \times 64$.

As shown in Fig. 10, compared to the original VC2, the 'VC2 + CV-VAE generates videos approximately four times longer, resulting in smoother motion. This further validates the feasibility of obtaining a compatible video VAE through latent regularization, thereby avoiding the massive computational power required to train a video diffusion model from scratch.

# B  Society Impacts

The CV-VAE can be seamlessly integrated into existing diffusion models, replacing the original 2D VAE for image or video generation, which may result in potential societal implications. While it proves beneficial in fields such as entertainment and advertising, by providing more realistic and immersive content, it also raises ethical and safety concerns. The ease of generating high-quality synthetic images and videos could lead to a surge in the production of harmful or misleading content, such as deepfakes, potentially exacerbating issues of misinformation and privacy invasion. We condemn the misuse of generative AI that harms individuals or spreads misinformation.

Real        Reconstructed        Real        Reconstructed

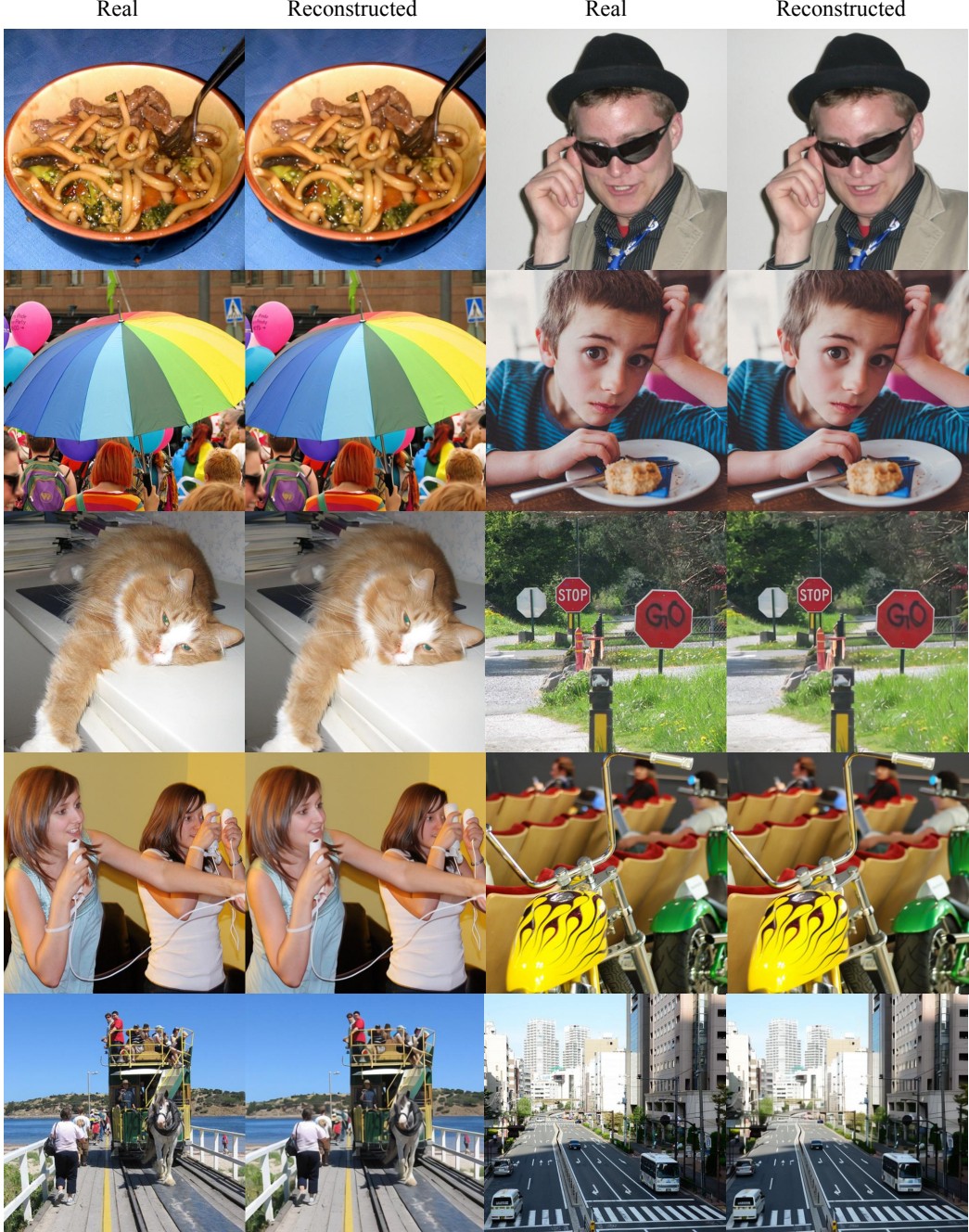

Figure 8: Our CV-VAE is capable of encoding and reconstructing images with high fidelity.

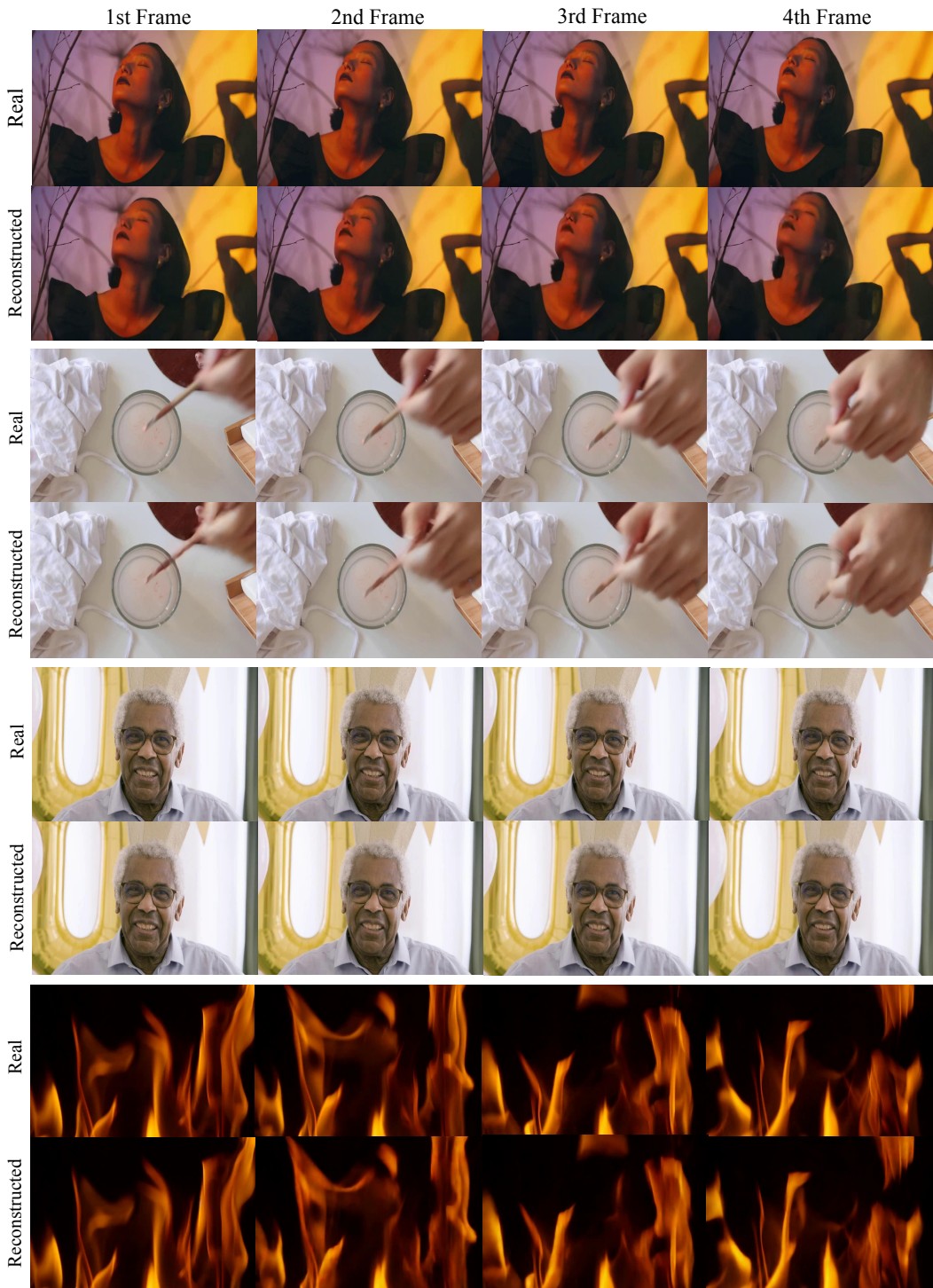

Figure 9: Reconstruction results of consecutive frames using CV-VAE.

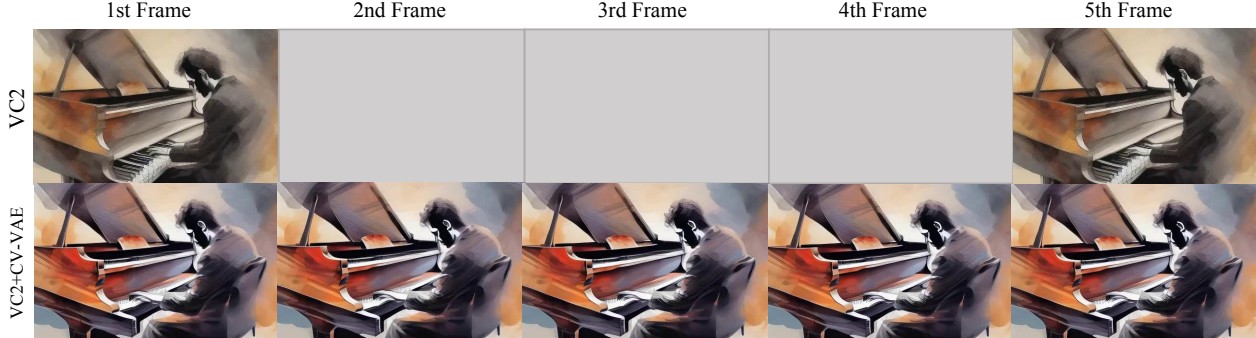

Prompt: pianist playing somber music, abstract style, non-representational, colors and shapes, expression of feelings, highly detailed

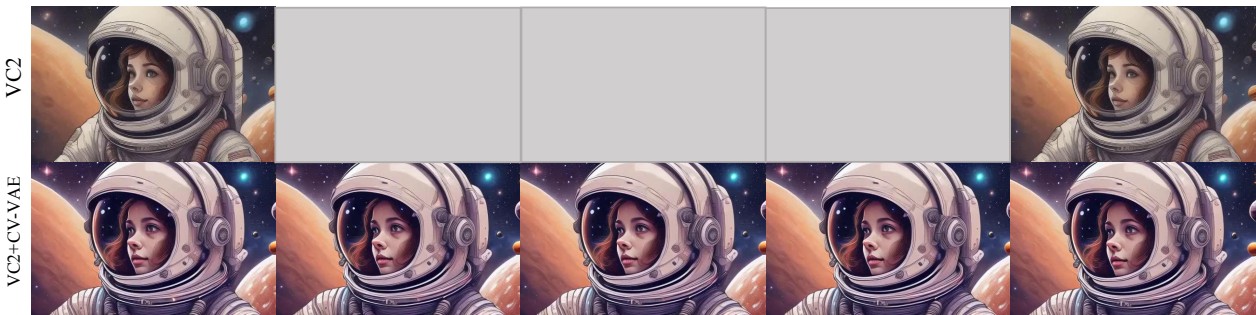

Prompt: a girl astronaut exploring the cosmos, floating among planets and stars, high quality detail

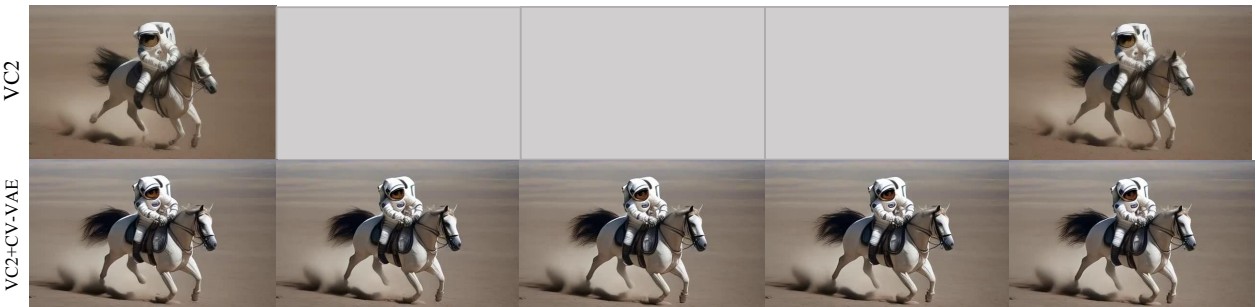

Prompt: A professional photograph of an astronaut riding a horse

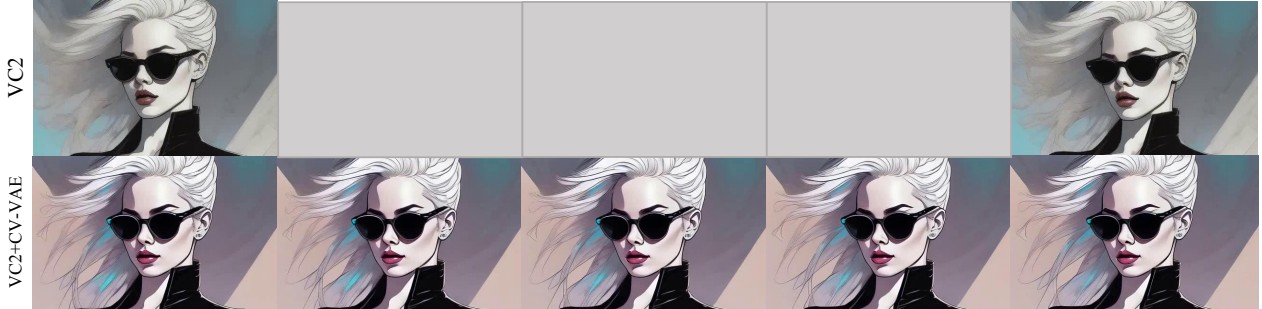

Prompt: black dress and sunglasses with white hair, in the style of becky cloonan, ross tran, dark white and cyan, close up, tsubasa nakai, light black and black, bold outlines, flat colors

Figure 10: Comparison between the image VAE and our video VAE on text-to-video generation of VC2 [7]. We fine-tuned the last layer of U-Net in VC2 to adapt it to CV-VAE. VC2 generates videos with a resolution of $16 \times 320 \times 512$, while the 'VC2 + CV-VAE' produces videos of $61 \times 320 \times 512$ resolution under the same computation. The missing frames in the VC2 results are marked in gray.

