# OpenReview forum: "CV-VAE: A Compatible Video VAE for Latent Generative Video Models"
_NeurIPS.cc/2024/Conference — NeurIPS 2024 poster_

### Official Review · Reviewer_LvuQ · 2024-07-09

**Soundness:** 3
**Presentation:** 3
**Contribution:** 3
**Rating:** 6
**Confidence:** 4

**Summary:**

The paper introduces CV-VAE, a 3D VAE that is trained through latent space regularization using an existing two-dimensional VAE decoder. This design facilitates seamless integration with 2D VAE-based diffusion models. The concept, while simple and straightforward, is notably efficacious and offers practical utility. Moreover, relative to conventional 2D VAEs, the proposed framework significantly compresses the latent space dimensionality by integrating temporal downsampling within the 3D VAE architecture.

**Strengths:**

1. The method is novel and delicate, which effectively compresses video latents and maintains latent distribution.

2. Experiments and visualizations demonstrate that CV-VAE performs equally to SD VAE on image-based/frame-based encoding and falls marginally behind on video encoding with little cost of finetuning SD models.

**Weaknesses:**

1. As presented in Figure 6, CV-VAE suffers from more severe flickering problems than the original SVD, which may possibly be due to the gap between the latent distribution of CV-VAE and SD-2.1. In the paper, the author also pointed out that text-to-image generation may experience differences in color. Authors may convince readers that such flaws can be overcome through further fine-tuning, e.g., fine-tuning full parameters of SVD, or fine-tuning for longer iterations.

2. Missing details about training: The authors didn’t present details about training batch size, augmentations, video fps, etc. Also, whether the default CV-VAE displayed in the experiment section is trained on video and image data separately or jointly (as vae of Opensora) remains unknown.

**Questions:**

1. Please refer to weakness.

2. In Table 3 and figure 6, authors may also present the metrics of SVD on 97 frames straightly or by using the frame interpolation model of SVD.

3. Authors may specify whether the decoder-based regularization is still employed when training with images as inputs for CV-VAE.

4. Authors may also present metrics comparison between SVD and SVD + CV-VAE on text-to-video generation.

5. Can the method be supported by theoretical proof? For example, can we minimize the distribution distance between video latents of CV-VAE and one of corresponding image latents encoded by 2d VAE?

---

> ### Author Rebuttal · Authors · 2024-08-06
>
> **W1: Flickering problems of CV-VAE**
>
> Our optimization loss is a trade-off between compatibility and better reconstruction quality. Therefore, there is still a domain gap between CV-VAE and diffusion models, which leads to color shifts or flickering problems. This offset can be alleviated by further fine-tuning.
>
> **W2: Missing details about training**
>
> Thank you for your comments. Actually, we provided the training details in the manuscript. (1) In the "Training Details" paragraph of Section 4.1, we introduced the training datasets, resolution, batch size, learning rate, and other augmentations. We also mentioned joint training with two image datasets of different resolutions (256x256, 512x512) and two video datasets of different resolutions (9x256x256, 17x192x192), with batch sizes of 8, 4, 1, and 1 for the four settings, respectively. (2) We will add some details about the video fps. During the training process, we randomly used a value between 1 and 4 as the frame stride for sampling.
>
> **Q2: Comparison between CV-VAE and interpolation model**
>
> Thank you for your suggestion. As SVD only released versions for 14 and 25 frames, direct inference of 97 frames would cause the video to collapse. Therefore, we provided comparative results for the frame interpolation model in attached file **(Table 3)**. We use RIFE [4] as the frame interpolation model for comparison, as it is popular and has 4.2K stars on GitHub. Our model outperforms RIFE in 2 out of 3 metrics, further validating the effectiveness of our method.
>
> **Q3: Decoder-based regularization during training with images**
>
> During the training process, we use both images and videos for joint training, so decoder-based regularization is also employed when taking images as inputs.
>
> **Q4: Comparison between SVD and SVD + CV-VAE on text-to-video generation**
>
> Thank you for your suggestion. Regrettably, SVD only released the image-to-video version. As an alternative, we compared the performance of VideoCrafter2 and VideoCrafter2+CV-VAE on text-to-video in Figure 10. Additionally, we provide the quantitative metrics and extra visual results for text-to-video in the attached file **(Table 4, Figure 2)**, which further validates the compatibility of CV-VAE with various video generation models.
>
> **Q5: Can the method be supported by theoretical proof?**
>
> Currently, the effectiveness of this method is proven through empirical evidence, and latent regularization can also be seen as a form of knowledge distillation [1], which is applied in different fields [2][3]. Providing a detailed theoretical proof is quite challenging and can be a direction for future exploration.
>
> [1] Distilling the knowledge in a neural network, arXiv 2015.
>
> [2] Adversarial diffusion distillation, arXiv 2023.
>
> [3] MiniLLM: Knowledge distillation of large language models, ICLR 2024.
>
> [4] Real-Time Intermediate Flow Estimation for Video Frame Interpolation, ECCV 2022.

---

> > ### Comment · Reviewer_LvuQ · 2024-08-14
> > **Response to Rebuttal**
> >
> > Thank you for your rebuttal and my concerns have been addressed. Accordingly, I have raised my rating to 6.

---

### Official Review · Reviewer_U7Gm · 2024-07-12

**Soundness:** 4
**Presentation:** 4
**Contribution:** 4
**Rating:** 7
**Confidence:** 5

**Summary:**

This paper designs a 3D VAE consistent with the 2D VAE, whose output mode can losslessly switch between 2D and 3D while retaining the characteristics of both the 2D VAE and 3D VAE. This allows for obtaining performance similar to the original 2D VAE while increasing the frame count of the 3D output.

**Strengths:**

1. Designing a 3D VAE with 2D latent consistent characteristics is very meaningful, and this paper successfully achieves this.

2. The bidirectional VAE alignment method designed in this paper is very clever. A similar design is found in [1], but the scenario used in this paper is completely different.

3. Achieving effective performance in both 2D and 3D simultaneously is of high practical value to the community, for example, supporting pre-training and frame interpolation in video Diffusion models.

4. The writing is fluent, and the experimental design is excellent.

[1] Christensen, Anders, et al. "Image-free classifier injection for zero-shot classification." Proceedings of the IEEE/CVF International Conference on Computer Vision. 2023.

**Weaknesses:**

1. Figure 10 attempts to demonstrate the effect of the 3D VAE on frame count enhancement. I understand that the conclusion of this figure is actually the most critical value proof of this paper: retaining the 2D VAE latent consistency while having the embedding capability of 3D features, such as extending the frame count. This is quite challenging, as the 3D VAE might tend to completely collapse to perform similarly to the original 2D VAE, simply replicating the 2D results. The loss between replicated 2D frames is very small, making optimization difficult. While the distance of the horse's legs in Figure 10 might prove it's not direct replication, the other two cases are hard to distinguish differences. Can the authors provide more evidence to explain that the model is not just learning simple replication?

2. In Table 1, when comparing, which pretrained 2D VAE is aligned with "Ours"?

**Questions:**

1. Would training the VAE with bf16 cause a performance collapse?

2. In Lines 218-219, how are the images of 256 × 256 and 512 × 512, as well as the videos of 9 × 256 × 256 and 17 × 192 × 192 organized during the training cycle? Do you train on images first, then videos, or start with low resolution and then move to high resolution?

3. What impact does training the VAE at low resolution or low frame rate have when performing SVD on high resolution and high frame rate?

**Limitations:**

Author added a limitation section in paper and I think there is no obvious potential negative societal impact.

---

> ### Author Rebuttal · Authors · 2024-08-06
>
> **W1： More examples to explain the model is not just learning simple replication**
>
> Thank you for your suggestion. **(1)** Due to the size limitation of the paper (50MB), we did not include the text-to-video results as videos in the pdf. The visual results were provied in the supplementary material. **(2)** We alslo provide more visual results of VideoCrafter2 + CV-VAE and quantitative metrics in the attached file **(Table 4, Figure 2)**. It can be observed that the motion between frames is smooth, and it is not simply a duplication.
>
> **W2:  Which pretrained 2D VAE is aligned with "Ours" in Table 1?**
>
> Thank you for your comments. In Table 1, both of our models (2D+3D, 3D) are aligned and compatible with VAE-SD2.1. Our models are also aligned with a lot of open-source diffusion models in the community that use VAE-SD2.1 as the auto-encoder.
>
> **Q1: Would training the VAE with bf16 cause a performance collapse?**
>
> Yes, training the model with either bf16 or fp16 would result in numerical collapse (NaN), which might be due to the instability caused by GAN loss. Therefore, CV-VAE is trained with fp32.
>
>
> **Q2:  Jointing training between images and videos**
>
> We use four settings for joint training: 256 × 256 images, 512x512 images, 9 × 256 × 256 video clips, and 17 × 192 × 192 video clips. In each iteration, a batch from one of the four settings is generated with different batch sizes (8, 2, 1, 1), allowing samples of different resolutions to be trained simultaneously.
>
>
> **Q3： Performing SVD on high resolution and high frame rate**
>
> Since CV-VAE is composed of convolutional networks and we train it on different resolutions and frame rates, CV-VAE can adapt to higher resolutions and different frame rates with a negligible performance degradation.

---

> > ### Comment · Reviewer_U7Gm · 2024-08-11
> > **Thanks for rebuttal.**
> >
> > I have no further questions, and I stand by my rating. Personally, I believe the quality and contribution of this paper generally meet the bar for NeurIPS.

---

### Official Review · Reviewer_tAbR · 2024-07-12

**Soundness:** 3
**Presentation:** 2
**Contribution:** 2
**Rating:** 5
**Confidence:** 5

**Summary:**

This paper proposes a new video VAE, starting with pretrained image VAE. It includes several techniques that are still capable of handling images and do not suffer from significant computational overhead. First, they use pretrained weights of 2D VAE (from Stable diffusion) by using their weights as initialization with model inflation. They also used efficient 3D architecture to only half of 2D convolutions as 3D convolutions to minimize the computation increase and also applied temporal tiling to handle long videos. They also use a pretrained frozen image decoder to regularize the latent space from the 3D encoder/encoder to ensure the latent space becomes similar to the stable diffusion latent space. The paper shows the proposed video encoder can have a compact latent space with a temporal compression factor (of 4) but is very similar to pretrained image latent space, which can be used for longer and smoother video generation by finetuning existing latent video diffusion models in this latent space.

**Strengths:**

- The paper is generally well-written and well-motivated. In particular, the paper tackles an important problem of constructing a compact latent space of videos that is similar to image space.
- The performance drop from fine-tuning does not seem that large and shows a good potential for training a latent video generation model.
- The paper shows a real use-case of this VAE by finetuning stable video diffusion.

**Weaknesses:**

- The paper lacks a comparison with recent video autoencoders. Specifically, the baselines that the paper provided are quite outdated (e.g., TATS and VQGAN), considering that there are many recent attempts to design a better video autoencoder, to name a few [1, 2, 3]. The authors should discuss what's the pros and cons of this approach compared with these approaches, and if possible, the authors should compare the performance as well. In particular, [2] provides very similar intuition to this paper because they also try to construct a latent space capable of jointly handling images and videos.
- Lack of novelty: The proposed method is mainly composed of components that have already been widely used to extend image models to videos.
- The paper does not mention the total training time (even though it includes the number of GPUs and the total number of training iterations).

[1] Video Probabilistic Diffusion Models in Projected Latent Space, CVPR 2023.
[2] Language Model Beats Diffusion: Tokenizer is key to visual generation, ICLR 2024.
[3] Efficient Video Diffusion Models via Content-Frame Motion-Latent Decomposition, ICLR 2024.

**Questions:**

- The paper only provides video generation results on image-to-video generation tasks; I wonder if one naturally achieves more smooth video generation model by fine-tuning existing text-to-video generation models in this latent space (such as modelscope).
- Are you planning to release the code and the model parameters?

**Limitations:**

The paper addresses the limitation appropriately.

---

> ### Author Rebuttal · Authors · 2024-08-06
>
> **W1: Comparison between recent video autoencoders**
>
> **(1)** There might be a missunderstanding. We have compared with the latest Video VAE in Table 1: VAE-OSP, which was released in May 2024. (2)  MAGVIT2 [2] and CMD [3] are SOTA methods, but they have not released their weights. Following the suggestion of the reviewer to compare with more recent Video VAEs, we add new quantitative and qualitative comparisons of Open-MAGVIT2 [4] (released in July 2024) and PVDM[1] in the attached file **(Table 2, Figure 1)**. Open-MAGVIT2 is an opensource project to replicate MAGVIT2.
>
>
> [1] Video Probabilistic Diffusion Models in Projected Latent Space, CVPR 2023.
>
> [2] Language Model Beats Diffusion: Tokenizer is key to visual generation, ICLR 2024.
>
> [3] Efficient Video Diffusion Models via Content-Frame Motion-Latent Decomposition, ICLR 2024.
>
> [4] Open-MAGVIT2: Democratizing Autoregressive Visual Generation, 2024.
>
>
> **W2: Lack of novelty**
>
> We would like to claim that the major contribution of this work is the framework (latent regularization with 2D decoder) to obtain a video VAE from a pretrained image VAE, which is compatible with existing image and video diffusion models, rather than a novel network component or a loss.  The compatibility greatly saves additional efforts of training diffusion models to adapt the VAE and being compatible with a wide range of community models. For example, Open-Sora-Plan, based on the incompatible Video VAE, spent an additional ~7138 GPU hours to obtain the image diffusion model.
>
>
> **W3: Training Time**
>
>  Training the CV-VAE took approximately 800 A100 GPU hours. We will clarify this in the final version.
>
>
>  **Q1: Results of Text-to-video diffusion model**
>
> Thanks for your concern. **(1)** We have provided the visual results of CV-VAE + VideoCrafter2 (text-to-video) in Figure 10 and the supplementary material. **(2)** We also provide the qualitative results and more visualizations in the attached file **(Table 4, Figure 2)**.
>
>
> **Q2： Release of code and weights**
>
> We will make the code and weights publicly available to promote community development. Following the rules of response, we have submitted a preview version of the open-source code to the Area Chair.

---

> > ### Comment · Reviewer_tAbR · 2024-08-11
> > **Response**
> >
> > Thanks for the clarifications. Many of my concerns are now addressed. Please add this discussion, particularly for a comparision with MAGVIT2 because it shares lots of similarities to this work. I have a few more questions:
> >
> > 1. What is a meaning of "Comp" in Table 1 in the rebuttal PDF file?
> > 2. I checked the MAGVIT2 repository and it seems there are two versions available - the first one uses 16x downsampling and the second uses 8x downsampling factor for training and inference. Which model is used for the comparison?

---

> ### Author Response · Authors · 2024-08-12
>
> Thank you for your comments. We clarify that CV-VAE and MAGVIT are completely different works: (1) MAGVIT2 represents images or videos as **discrete tokens**, which are used in **autoregression models** to generate images or videos; Our CV-VAE represents images or videos as **continuous latents**, which are used in **diffusion models** to generate images or videos. (2) Our main goal and contribution is to design a Video VAE that is compatible with 2D VAEs, while MAGVIT2 is not compatible with any other VAEs. here are the responses:
>
> **R1:** The term "Comp." has the same meaning as in Table 1 of the main paper, indicating compatibility. This also represents the core contribution of our work, which is to design a Video VAE that is compatible with other 2D VAEs, Image Diffusion Models, and Video Diffusion Models.
>
> **R2:** We use the 8x downsampling version from MAGVIT2 for comparison, which is also the same as the downsampling factor in our CV-VAE.

---

> ### Comment · Reviewer_tAbR · 2024-08-12
> **Response**
>
> Thanks for the further clarification - can authors can provide rFID and rFVD (reconstruction FID and FVD) of the proposed methods and other baselines in MSR-VTT? It seems MAGVIT-2 results show too high LPIPS and low SSIM scores considering that Open-MAGVIT2 shows quite a good performance on ImageNet and the original MAGVIT2 paper shows surprising low LPIPS on UCF-101.

---

> ### Author Response · Authors · 2024-08-12
>
> The comparison results of FID and FVD metrics on MSR-VTT are as follows. we use the first frame of MSR-VTT to calculate FID.
>
>
> | Methods      | Comp.      | FID  | FVD   |
> |:--------------:|:------------:|:------:|:-------:|
> | VAE-SD2.1    | -          | 1.31 | **7.42**  |
> | VQGAN        | $\times$   | 7.56 | 23.59 |
> | TATS         | $\times$   | 8.27 | 19.36 |
> | VAE-OSP      | $\times$   | 1.28 | 9.81  |
> | PVDM         |$\times$    | 7.74 | 22.38 |
> | Open-MAGVIT2 | $\times$   | 2.38 | 12.13 |
> | Ours         |$\checkmark$| **1.26** | 8.55  |
>
> Due to quantization error, discrete VAEs usually lose more information than continuous VAEs. For example, MAGVIT2 encodes an image with resolution of $256\times 256$ into **integers** ranging from 0 to 262144 with a size of **$32\times 32$**, while continuous VAEs (SD-VAE2.1, VAE-OSP, CV-VAE) encode an image with resolution of $256\times 256$ into **floating-point vectors** with a size of **$32\times32\times z$** where $z$ represents channels of latent and are equal to 4 for the continuous VAEs in Table 1. Therefore, it is not surprising that continuous VAEs can achieve better reconstruction results than discrete VAEs.
>
> Moreover, reconstruction quality is not our primary goal in designing CV-VAE, since the reconstruction quality can be easily improved by increasing the size of $z$ without changing the model structure and size [1][2]. For example, in Stable Diffusion 3[2], the PSNR of VAE with $z = 4$ is 25.12dB, while the PSNR of VAE with $z=16$ is 28.62dB. Our primary goal is to design a compatible Video VAE based on the existing 2D VAE, so we must keep the size of $z$ the same as the 2D VAE ($z=4$ for VAE-SD2.1).
>
>
> [1] Emu: Enhancing Image Generation Models Using Photogenic Needles in a Haystack, arXiv 2023.
>
> [2] Scaling Rectified Flow Transformers for High-Resolution Image Synthesis, ICML 2024.

---

> > ### Comment · Reviewer_tAbR · 2024-08-13
> > **Response**
> >
> > Thanks for the further clarification. I increased my rating from 4 to 5 accordingly.

---

### Official Review · Reviewer_xmxT · 2024-07-16

**Soundness:** 3
**Presentation:** 3
**Contribution:** 2
**Rating:** 6
**Confidence:** 4

**Summary:**

This paper focuses on a compact video VAE suitable for both image and video generation tasks. The motivation stems from the absence of spatial-temporal compressed continuous latent 3D VAE models for video generation. To effectively utilize 2D VAE and seamlessly integrate image generation models, the authors propose a latent space regularization and initialization method to maximize the use of 2D VAE. A 3D CNN-based architecture is also proposed to compress both temporal and spatial dimensions efficiently. The evaluation of the proposed method is primarily based on reconstruction tasks and image/video generation tasks.

**Strengths:**

The motivation is clear: the current video generation method may suffer from a lack of high-quality and high-efficiency spatial-temporal VAE if it can be used with the existing T2I model, which also makes it more general and effective.
The decoder loss seems a simple and novel design, which helps the model learn spatial latent efficiently and align with the original 2D VAE. Initialization of 3D VAE  with 2D ones seems to provide some insights in practice.
The qualitative results are good.

**Weaknesses:**

1. Improvements over the baseline are presented in Table 1, where the authors compare the main reconstruction results with different open-sourced VAE methods. However, I don’t think it is a fair comparison to claim superior performance. The training data is a key factor in this table. The authors use in-domain data, such as Webvid, for both training and evaluation, whereas other methods do not. Therefore, it is unclear if the proposed method can truly outperform other methods when trained on the same data.

2. Many generation models have been released based on SD VAE. Since compactness does not require further fine-tuning, it is crucial to evaluate more models to strengthen the claim. Evaluating only one model in Table 2/3 is not convincing.

**Questions:**

The term "mapping function" seems to require clarification, as it appears to be more of a simple sampling process from video frames. In Table 7, a higher CLIP score is better. Why does random sampling work best? Providing more exploration and explanation can help readers understand this better.

**Limitations:**

Yes, the author discussed the limitations and potential social impact.

---

> ### Author Rebuttal · Authors · 2024-08-06
>
> **W1:  Fairness of training and evaluation data**
>
> Thank you for your comments on the fairness. **(1)** Unified training data. Since the VAE models in Table 1 are trained on different datasets and some of them did not release their training and dataset details, such as VAE-OSP, it is difficult to train all models on the same dataset. **(2)** Unified testing data. We appreciate you pointing out that evaluating on the in-domain validation set is unfair. Therefore, we conduct additional evaluations of all VAE models on the publicly available MSR-VTT validation set that is not mentioned as the training set of those models. The evaluation results are presented in the attached file **(Table 1)**. Our method performs better than other video VAE models.
>
> **W2: Evaluation on different diffusion models**
>
> **(1)** We would like to first clarify that we have performed compatibility validation on multiple diffusion models, including quantitative evaluations on SD2.1 (text-to-image), SVD (image-to-video) (Table 1, 2), and visual evaluations on SD2.1, SVD, and Videocrafter2 (text-to-video) (Figure 5, 6, 10). **(2)** To further validate our compatibility with text-to-video models, we have additionally conducted quantitative evaluations on Videocrafter2+CV-VAE and also provided visual results in the attached file **(Table 4, Figure 2)**.
>
> **Q1:  Clarification of "mapping function"**
>
> Thank you for your comments. We will revise "mapping function" to "sampling function" to clarify the expression. We observed that the "1st Frame" results in poorer reconstruction of subsequent frames due to the lack of constraints on other frames, "Average" leads to more motion blur in the reconstructed frames, and "Slice" causes frames without 2D Decoder constraints to be more prone to artifacts. "Random" is proposed based on "Slice" to cover the content in the whole sequence and uses the randomly sampled frames instead of the average frame in a slice to avoid motion blur.

---

> > ### Comment · Reviewer_xmxT · 2024-08-12
> > **Thanks for the rebuttal**
> >
> > The additional results addressed my concerns.

---

### Author Rebuttal · Authors · 2024-08-06

We thank the reviewers for their thoughtful analysis and feedback, we are glad the reviewers find that

* Ours proposed question are valuable
    * "Designing a 3D VAE with 2D latent consistent characteristics is very meaningful, and this paper successfully achieves this " - Reviewer U7Gm.
    * "Achieving effective performance in both 2D and 3D simultaneously is of high practical value to the community" - Reviewer U7Gm.
    * "the paper tackles an important problem of constructing a compact latent space of videos that is similar to image space." - Reviewer tAbR.

* Our solutions are novel and effective and our experiments are well-conducted
    * "The bidirectional VAE alignment method designed in this paper is very clever." - Reviewer U7Gm.
    * "The method is novel and delicate, which effectively compresses video latents and maintains latent distribution." - Reviewer LvuQ.
    * "The decoder loss seems a simple and novel design, The qualitative results are good." - Reviewer xmxT.

* Our paper is well-written
    * "The writing is fluent" - Reviewer U7Gm.
    * "The paper is generally well-written" - Reviewer tAbR.

Attached you can find a file containing new experiments suggested by the reviewers.

*  Reconstruction comparison results of different VAEs on a public out-domain dataset.
*  Quantitative and visual comparison results of CV-VAE with some recent methods.
*  Quantitative and visual results of CV-VAE in text-to-video generation models.
*  Comparison results between CV-VAE and interpolation model.

---

### Decision · Program_Chairs · 2024-09-25

**Decision:**

Accept (poster)

**Comment:**

The paper is reviewed by four expert reviewers and got overall postiive ratings of accept, two weak accepts and one borderline accepts. Reviewers find that the paper tackles an important problem of desiging 3D VAE for compact video representation and well executed. Some reviewers concerns regarding the comparison to more recent works, lack of technical novelty, have been addressed during rebuttal. AC agrees that the paper provides a good insight on building compact video representation and recommend for acceptance.